# New Facility for Membrane Fouling Investigations under Customizable Hydrodynamics: Validation and Preliminary Experiments with Pulsating Cross-Flow

**DOI:** 10.3390/membranes12030334

**Published:** 2022-03-17

**Authors:** Roberto Bert, Costantino Manes, Alberto Tiraferri

**Affiliations:** 1Department of Environmental Land and Infrastructure Engineering (DIATI), Politecnico di Torino, Corso Duca degli Abruzzi 24, 10129 Torino, Italy; costantino.manes@polito.it (C.M.); alberto.tiraferri@polito.it (A.T.); 2CleanWaterCenter@PoliTo, Politecnico di Torino, Corso Duca degli Abruzzi 24, 10129 Torino, Italy

**Keywords:** membranes, ultrafiltration, fouling, fouling reduction, filtration rate, channel flow, pulsating flow, PIV

## Abstract

Flux reduction induced by fouling is arguably the most adverse phenomenon in membrane-based separation systems. In this respect, many laboratory-scale filtration studies have shown that an appropriate use of hydrodynamic perturbations can improve both performance and durability of the membrane; however, to fully understand and hence appropriately exploit such effects, it is necessary to understand the underpinning flow processes. Towards this end, in this work we propose and validate a new module-scale laboratory facility with the aim of investigating, at very well-controlled flow conditions, how hydrodynamics affects mass transport phenomena at the feed/membrane interface. The proposed facility was designed to obtain a fully developed and uniform flow inside the test section and to impose both steady and pulsating flow conditions. The walls of the facility were made transparent to grant optical accessibility to the flow. In this paper, we discuss data coming from particle image velocimetry (PIV) measurements and preliminary ultrafiltration tests both under steady and pulsating flow conditions. PIV data indicate that the proposed facility allows for excellent flow control from a purely hydrodynamic standpoint. Results from filtration tests provide promising results pointing towards pulsating flows as a viable technique to reduce fouling in membrane systems.

## 1. Introduction

Water management is a highly topical issue and will be even more so in the near future. According to the United Nations Water Resources Report 2021, the use of freshwater has increased six-fold in the last 100 years and demand continues to grow by 1% per year [1]. In this perspective, desalination plants offer an economically viable solution for the production of drinking water. Starting from 326 m^3^/day produced worldwide in 1945, the use of this technology based on membrane systems has increased exponentially to over 80 million m^3^/day in 2013 [2,3] and even more today. Moreover, thanks to their versatility, membranes have also found increasing use in other processes related to the food, medical and chemical sectors and more generally in the treatment of wastewater from industrial plants [4,5,6]. In relation to the importance acquired, due to the vastness of the possible uses, there has also been a natural expansion of the research-related sector with a strong increase in the number of studies and publications [7] relating to process optimization and new applications.

Although technological development [8,9] and process intensification efforts have improved membrane separation processes significantly, fouling still represents a major issue, as it causes a decrease in productivity resulting from the accumulation of particles, microorganisms and macromolecules on the membrane surface with consequent negative outcomes in terms of permeate quality and increased energy consumption. Mitigating such fouling effects essentially means finding ways to minimize deposition at the fluid–membrane interface while granting a reasonable amount of filtrate flux. This may be achieved by maximizing mixing in the feed while minimizing energy expenditure to generate it. Towards this end, the so called “hydrodynamic methods” offer an interesting opportunity. In a recent review paper, Jaffrin [10] (and references therein) provides an appraisal of all such methods and highlights that an appropriate use of hydrodynamic perturbations can improve the performance of membrane filtration at low energetic costs quite significantly. For example, in addition to the classical retro-filtration approach, applications of helicoidal flows, vibrating and rotating systems and pulsating flows are reported and deemed to be promising.

While the potential of these techniques is not in doubt, the physics underpinning the processes leading to the observed reduction in fouling is not well understood and results provided by the literature are often presented in dimensional form [11,12,13,14], which makes them have no general validity and only a (weak) support for an empirical, and hence unreliable, design of anti-fouling hydrodynamic techniques.

The aim of the present paper is to propose, test and validate a novel experimental facility designed and built with the intention of providing data that will help bridge this knowledge gap. The proposed facility was designed considering similarity principles and allowing for state-of-the-art laser diagnostics techniques to measure flow velocities and mass transfer process occurring at the membrane-feed interface. Aside from steady flows, the current version of the experimental set-up allows for the generation of pulsating flows, which are identified by the authors as a hydrodynamic method that combines a good potential for fouling reduction with ease of implementation and installation in industrial applications. Some preliminary results are presented in support of this hypothesis.

The paper is organized as follows: Section 2 is devoted to the accurate description of the facility as well as its design rationale. Section 3 describes the equipment and experimental procedures adopted in the tests. Section 4 presents results with a view to demonstrate that: (i) the facility was well-designed from a fluid-dynamics stand-point and allows for the generation of nicely controllable canonical channel flows that can be investigated by means of laser diagnostic techniques; (ii) filtration tests are responsive to changes in flow conditions and provide repeatable and encouraging results especially for what concerns pulsating flow conditions. Section 5 presents overall conclusions from the present study.

## 2. Experimental Methods

### 2.1. Overall Description of the Facility

The hydraulic plant shown in Figure 1a, mainly made in stainless steel 316L to ensure durability, is a recirculating facility whereby the feed is drawn from a 1 m^3^ (constant-head) main tank by a pumping system consisting of two in-parallel multistage pumps, with a maximum total flow rate of 2.4 L/s. The two pumps are mounted on two separated pipe-branches equipped with in-line ball valves for sectioning the circuit and imposing a wide range of flow rates to cover laminar, transitional and turbulent flow regimes in the test section. Just further downstream, the system splits into a main pipeline and a by-pass line that returns a selected fraction of the feed directly to the reservoir. Along the former, after a few meters, a divergent allows for connection to a flow conditioning system made of a settling chamber and a contraction that minimizes incoming turbulence and allows for the undisturbed development of boundary layers within the test section (the flow conditioning section is described in detail below). At the downstream end of this test section, a sensor records hydraulic pressure in real-time and a short stretch of pipe closes the hydraulic circuit. Where possible, small diameters have been used, partly to reduce the total volume and partly to better control the flow rate in the main line. Details about pipe diameters, tanks capacity (i.e., both the main tank and the collecting thanks), test section and pump types with three points of the flow vs. pressure curve are provided in Appendix A.

The desired conditions in terms of pressure and flow rate are achieved through the combined regulation of pump frequency and control devices. A computer-programmable inverter allows us to fix the rotational speed of the pump shaft. The two globe valves on the delivery line are used to set the amount of water in each branch and one valve on the return line, downstream of the test section, acts as a back-pressure valve. Two self-closing air release valves are then installed at the highest elevations of the pressurized part of the circuit to allow for air removal. Right below the test section, three small trays collect the filtrate. 

Their emptying is regulated automatically by ultrasonic level sensors that control the on/off sequences of the bilge pumps. The filtrate then returns, via recirculation tubes, to the main tank to maintain a constant water volume in the whole system. The overhead tank is equipped with a thermocouple that allows for the monitoring of water temperature. An ultrasonic flow meter is mounted on the main line and a pressure sensor at the end of the test section. These were chosen with an appropriate level of accuracy, namely, 2% for u > 0.3 m/s for the AcquaTrans™ AT600 flow meter manufactured by Panametrics, and 0.008 bar for the Wika P-30 pressure sensor. Flow rate and pressure data, acquired at whatever desired frequency, are transmitted with a 4–20 mA signal to a National Instruments data acquisition card by means of a Labview program. To avoid changes in water density and in membrane hydraulic conductivity, temperature is kept constant by means of a 5 kW chiller. For a better and more detailed understanding of the overall system operation, reference should be made to the “Piping and instrumentation diagram” (P&ID) available in Appendix A.

### 2.2. Description of the Test Section

The main part of the system is the test section (test section is often used as fluid dynamicists’ jargon, while the community pertaining to membrane science often refers to it as housing cell or simply cell; all these names are herein used interchangeably), which houses the flat-sheet membrane. The section consists of a rectangular channel (1.45 m long, 0.2 m wide and 0.01 m high) made entirely of transparent plexiglass, which allows for optical access to cameras and lasers, hence assessment of membrane surface alteration and velocity measurements at any location (e.g., for measurements by means of particle image velocimetry, PIV, as presented herein).

Figure 1b shows the configuration of the housing cell hosting an impermeable smooth surface that was used for checking flow development under canonical channel flow conditions and as a benchmark for comparison with flows developing over porous membranes. Figure 1c provides details of the cells when equipped with the flat-sheet membrane. The upper wall of the housing is the same as in Figure 1b. The lower part is made of a rigid stainless steel porous medium (3.175 mm thick, 316LSS porous medium with uniformly distributed 10 μm pores) providing support for different types of flat-sheet membrane. The medium leans over a plexiglass bed that was subdivided into three separate compartments to collect the filtrate from different locations along the direction of the flow and hence to check for spatial uniformity of filtration rates during an experiment. Towards this end, the bed of each compartment has five bottom holes connected to drainage pipes that convey water to the underlying trays where level meters allow for the monitoring of filtrated water volumes over time and hence of filtrate flow rates as sought.

The design of the test section was carried out to accommodate the following requirements:(i)The development of a so-called “canonical” channel flow (i.e., a rectangular channel flow as largely investigated within the remit of the literature pertaining to fluid dynamics) allowing for full control and extensive validation of flow properties. Following the recommendations from the literature, the channel was designed with a large aspect ratio (i.e., ratio between width and depth) of 20, which allows flow properties in the mid cross section to be independent of lateral walls and hence comparable to theoretical predictions as well as measured or numerically modeled data [15];(ii)The length, L, of the channel had to be minimized to avoid working with very large volumes of water while allowing for fully developed and self-similar flows (i.e., flows whose velocity statistics, once appropriately scaled, are independent of the longitudinal coordinate) to form. This put constraints on the height H of the channel, because the higher the H the longer is the required L to obtain fully developed flows [16].

Requirements (i) and (ii) were nicely met with the test section having the following geometry: L = 1.45 m, H = 0.01 m and width B = 0.2 m. It is worth mentioning that the proposed facility was designed to investigate, at a fundamental level, how mass transfer processes occurring at the feed–membrane interface respond to a range of imposed (and well controllable) flow conditions, and not to replicate geometric features as encountered in industrial applications. The latter goal would indeed require working with complex geometries (see, e.g., spiral wound or pressurized hollow-fiber modules) that would make this job extremely challenging. Nevertheless, it was deemed necessary to design the test section to allow for dynamic and kinematic similarity with industrial flows. To the authors’ opinion, the two non-dimensional numbers that dictate the achievement of the sought similarity are the already mentioned Reynolds number, Re = Uh/ν, and the so-called suction coefficient, Γ = v/U [17], where v is the vertical velocity at the feed–membrane interface and U is the bulk mean velocity of the overlying channel flow. Whenever these two numbers have each an equivalent value in different systems (e.g., our lab setup and an industrial setup, or two lab setups), the systems may be considered equivalent from a hydrodynamics standpoint. The chosen system of pipes and test section geometry permits to impose a wide range of feed–flow velocities, U, and internal pressures to be explored. These conditions result in the possibility to impose a wide range of Re (1000 ÷ 20,000) and achieve a large range of Γ (covering up to two orders of magnitude) in relation to the permeability of the installed membrane sheet and applied pressure. Indeed, the proposed setup nicely covers typical hydrodynamics conditions used in industrial settings spanning from microfiltration to nanofiltration applications.

### 2.3. Description of the Flow-Conditioning Unit and Pulsating Flow System

Much of the quality of the flows that develop in the test section depends on the reliability and functioning of the flow conditioning unit, which is described in detail as follows. The abrupt widening of the settling chamber is designed to slow the fluid down so that quasi-irrotational motion can be established. Further downstream, the fluid gradually re-accelerates while flowing into the ad hoc designed convergent that allows for further turbulence dissipation through the imposition of a strong favorable pressure gradient, which is known to be effective for flow laminarization [18,19]. The design of the flow conditioning unit was a non-trivial task and driven by the need to meet several requirements. In particular:(i)The contraction ratio (the ratio between the cross-sectional area of the settling chamber and the test section) had to be at least equal to 4 for an effective turbulence damping to take place [20,21];(ii)The convergent had to be gradual to avoid boundary layer separation and the potential shedding of undesired eddies resulting from boundary layer separation;(iii)The convergent shape was very unconventional (i.e., never investigated before) as it had to gradually join the bulk circular cross section of the settling chamber to the wide and thin rectangular slit of the test section.

To meet all these requirements, the entire conditioning unit was carefully designed with the aid of computational fluid dynamics (CFD) and various numerical simulations were carried out assuming laminar flow conditions to develop in the unit (see an example in Figure 2).

Results from the simulations indicated that the chosen geometry allowed for smooth flow acceleration and a laterally uniform flow developing within the test section. Concerning the generation of pulsating flows, Figure 1d illustrates the proposed device which is based on the work published by Ramaprian and Tu [22]. A rod with a 45° angle cut was inserted into a 90° elbow and connected to a 0.37 kW electric motor equipped with speed drivers and gears. By adjusting the rotation speed, rs(5 ÷ 143 RPM), and the distance, *k*, of the rod with respect to the elbow, the ratio between the outflow area and the pipe diameter can be varied harmonically and a pulsating flow established in the system.

## 3. Experimental Procedure and Measuring Equipment

This section describes the measuring systems and procedures used to carry out the experiments. The section is divided in two parts. Section 3.1 is devoted to the description of the (herein called) fluid dynamic tests, which were carried out with the unit equipped with an impermeable plexiglass floor with the goal to test the development of canonical channel flows. Section 3.2. reports how filtration tests were carried out and describes the experimental technique used to monitor membrane fouling.

### 3.1. Fluid Dynamic Tests

Velocity measurements were carried out by means of a PIV system. This technique employs a series of image pairs, acquired at predefined time intervals, to evaluate the displacement of individual seeding particles illuminated by a planar laser sheet. Image pairs are compared using a cross-correlation algorithm, providing a two-dimensional velocity field with a spatial accuracy of 16 × 16 pixels, herein corresponding to approximately 0.19 × 0.19 mm. The equipment used in this study is a PIV Dantec system consisting of a 200 mJ YAG laser (wavelength of 532 nm corresponding to green light) that can be shot at a maximum frequency of 15 Hz. Images were acquired by means of an IMPEREX c-4080 camera (sensor size 18.8 × 14.1 mm) coupled to a 200 mm Nikon lens. The laser beam is conveyed through an optic-fiber cable into a mechanical arm ending with an optical unit made of cylindrical and spherical lens that allow for laser sheet generation and control in terms of width and focusing. Both optical unit and camera were mounted on a structure of modular aluminum profiles that allowed for very precise positioning and control of the alignment and parallelism between the camera, the main flow direction and the laser plane, whose thickness in the present experiments was set between 1 and 2 mm. Acquired images were then calibrated using a custom-built calibration plate inserted within the unit and then analyzed with DynamicStudio 6.4 to work out the velocity field.

Measurements aiming at checking flow uniformity within the test section were taken at three different longitudinal positions and a fixed distance from the channel side wall (i.e., z = 5 cm). In particular, with reference to the coordinate system presented in (Figure 1b) the laser sheet was first placed in the cross section identified by longitudinal coordinate x = 82.5, then at an upstream (x = 34.5) and finally at a downstream cross section (x = 130.5), respectively. Each of the three PIV acquisitions was conducted at the same dimensionless Reynolds number Re = Uh/ν = 4000 (U is the mean velocity, h is the semi-height of the channel and ν is the kinematic viscosity) or equivalently, Re* = u*h/ν = 70 (u* is the friction velocity defined as u* =$\sqrt{\tau_{0}/\rho }$, where ρ is the fluid density and $\tau_{0}$ is the wall shear stress, namely, the friction forces per unit area acting at the bottom wall), to achieve dynamic similarity with flows typically established in traditional membrane systems [23,24]. Since at this Re the flow is turbulent, 2000 image pairs were acquired, in order to estimate both first and second-order velocity statistics, with good confidence. A summary of experimental acquisitions is given in Table 1 in which we want to point out that the small differences in the Reynolds numbers are attributable to experimental variability.

### 3.2. Filtration Tests

Experimental findings were obtained from pulsating and steady-state flow tests. In both cases, 5 g of kaolin ($D_{50}$ = 1.5 μm) and 10 g of green clay ($D_{50}$ = 0.2 μm), which represent the main colloidal foulants in this application, were added to 555 l of tap water (feed concentration c = 27 ppm and feed viscosity ν = $10^{-6}$ m^2^/s), in order to reproduce the same total suspended solids (TSS) concentrations and similar particle composition as encountered in a nearby river (Po River [25,26]), which is currently used as main source of water for the city of Turin, and whose water will be soon potabilized by means of a new ultrafiltration plant, which is currently being installed by the water provider. A new polyethersulfone (PES) Synder-LY ultrafiltration flat-sheet membrane with a molecular weight cut-off (MWCO) of 100,000 Da and a nominal permeance between 80 and 123 $L/m^{2}/h/\mathrm{bar}$ was mounted in the housing section for each test. An average Re of 4000 was maintained in all tests with a constant fluid temperature (in the range of 18 ÷ 20 °C) and the applied pressure, *P*, was chosen between 1 and 1.5 bar. At these conditions, the suction parameter Γ varied between 1.37 × $10^{-4}$ and 1.84 × $10^{-4}$. In order to also obtain preliminary data about potential fouling reduction under pulsating flow conditions, one experiment was carried out under the same hydraulic conditions as per the steady filtration tests (i.e., *Re* = 4000 and Γ = 1.37 ÷ 1.84 × $10^{-4}$) but, this time, forcing the flow rate and pressure to undergo oscillations induced by the pulsed flow system described in Section 2.3 (see also Figure 1d). Towards this end, the distance between the rod and the elbow, *k*, was set to 13.3 cm and the rotation frequency equal to 0.09 Hz. This corresponds to variations of pressure and flow rate contained between 1.51 ± 0.04 and 0.42 ± 0.035, respectively. The frequency of the imposed oscillations sets the dimensionless frequency $\omega^{+}=2\pi$ f ν/u*^2^ to be equal to 0.0025, which, according to the literature pertaining to turbulent pulsating flows, corresponds to a regime whereby turbulence time scales are much smaller than the period, 1/f, of the oscillations [27]. In terms of particle mass transport, it was hypothesized that this condition might be favorable to diminish particle depositions with respect to a steady flow condition. This because the high shear associated with the acceleration phase of an oscillation has the time to develop into enhanced turbulence (i.e., enhanced mixing) before it is damped by the decelerating phase. This “high-intensity” turbulent bursts should, in principle, increase the probability of particles to be entrained in the feed rather than depositing over the membrane surface. A summary of experimental conditions is given in Table 2.

In order to obtain a significant and measurable drop in the filtration rate over time, each test was carried out for a duration of 23 h. During all the filtration tests, the surface of the membrane was monitored by a series of photographs taken at 15 min intervals using a camera (Canon EOS 700) controlled by digiCamControl software. The camera was carefully mounted horizontally (i.e., parallel to the membrane) and was placed above the test section at a sufficient distance to capture its entire extent. A dark room was built around this portion of the system and 11 spotlights of 600 LM each were placed inside it in a fixed position to ensure uniform and constant light during the tests. An initial white balance was carried out for all the images and then the variation of the yellow band (which is the dominant one in the measured spectrum) was calculated with respect to the initial value of the corresponding area, providing the % of color variation in time at different locations over the membrane. This system allows to monitor the chromatic change of the membrane surface induced by colloidal fouling in time and, even more interestingly, in space to check for particle deposition uniformity over the entire cell unit.

## 4. Results

This section is split in two parts. Section 4.1 and Section 4.2 provide results obtained from the fluid dynamic and filtration tests, respectively.

### 4.1. Fluid Dynamic Tests

This section contains the experimental results of the fluid dynamic tests carried out in the three different longitudinal positions within the test section, whose parameters are reported in Table 1. Figure 3a shows the mean flow field, superimposed to a PIV snapshot and the contour plot of the longitudinal velocity obtained from the time averaging of 2000 flow resolutions, obtained from PIV in the F-1 test.

The contour plot shows quite uniform flow conditions, thus indicating that the laser was positioned correctly and parallel to the flow. This gives confidence about measurements quality when assessing the standard “family portrait” of velocity statistics profiles reported in the other panels of the figure. These were obtained by further averaging flow properties obtained locally (i.e., for each PIV interrogation areas) over lines parallel to the wall to achieve enhanced statistical robustness. The vertical coordinate and the velocity statistics profiles presented in panels (b–f) of Figure 3 are plotted following a conventional normalization, employing the channel half-height, h, the maximum speed, $U_{m}$, and the friction velocity, u*=$\sqrt{\tau_{0}/\rho }$ ($\tau_{0}$ is the wall shear stress and ρ is the water density). As is common in turbulent channel flows, $\tau_{0}$ was estimated, for each of the three cases, from the linear extrapolation to the bottom wall of the total shear stresses, τ, defined as $\tau \left(y\right)=\rho \nu \frac{d\bar{u}}{\mathrm{dy}}-\rho \bar{u^{\prime }v^{\prime }}$, where ν is the kinematic viscosity, u is the longitudinal velocity component, $u^{\prime }$ and $v^{\prime }$ are the longitudinal and vertical velocity (time) fluctuations, respectively, and overbar denotes time-averaging [28]. Results from the extrapolation are reported in Figure 3b, which shows how profiles of τ as obtained from PIV measurements are reasonably linear as expected in uniform turbulent channel flows.

Figure 3c–f demonstrate that all the measured vertical profiles of mean velocities (panels (c) and (d)) and velocity variance (panel (e)) and covariance (panel (f)) collapse well, thus indicating the flow to be self-similar. Moreover, since all the scaling parameters used in Figure 3 are reasonably constant for each test (i.e., at each different longitudinal coordinate) the flow can be considered uniform and hence fully developed. Further support about the quality of the data comes from the comparison with the direct numerical simulations of Tsukara et al. [29], performed at similar $Re^{*}$, whose data collapse very well with those presented herein, thus further demonstrating that the flows conform to canonical, turbulent channel flows, as sought. Further evidence for the occurrence of fully developed turbulence (and not transitional behavior) can be found in the Appendix A where, for both velocity fluctuations u${}^{\prime }$ (longitudinal) and $v^{\prime }$ (wall-normal), a time series at a fixed point in space (Appendix A) and a contour plot within the measurement window (Appendix A) are presented. Both figures show that the flow presents a chaotic and irregular continuous pattern without the occurrence of intermittent laminar vs. turbulent behavior that is typically encountered in transitional conditions.

### 4.2. Filtration Tests

Figure 4 provides results of test S-4_1.5, which was conducted at steady flow conditions and an applied pressure of 1.5 bar.

The filtrate corresponding to each third of the membrane area, identified as “up”, “mid” and “down”, along the longitudinal direction of motion, is conveyed into the respective collection tray, which is then automatically emptied when a predefined value is reached (Figure 4a). The integral of these values, for each of the three areas, gives the respective cumulate from used to derive the flux associated with the test, as illustrated by Figure 4b. Since the global slope of the curves related to each section is almost the same, it is possible to infer that there is no evident longitudinal spatial variation in the filtrate flux. Note that in the filtration tests, part of the discharge is progressively lost along the direction of motion through the membrane (in our case a value between 1.9% and 2.5%), hence a slight non-uniformity of flow conditions should be expected. Our results indicate that such a slight non-uniformity has a negligible effect on the bulk flux of filtrate, which is essentially the same for the up, mid and down sections.

More spatially resolved information about the uniformity of the fouling rate is provided by the analysis of the photographs taken with the overhead camera, which is presented in figure Figure 5 for one experimental condition (the other tests showed a similar behavior as can be seen from Appendix A).

Here, panel (a) reports examples of the images taken over different time steps (note that full optical access to the surface of the membrane is prevented due to the presence of braces that were used to avoid water leakage), while panel (b) shows the variation in yellow band color as detected by the camera over the 11 boxes highlighted in red in panel (a).

Panel (b) indicates that differences in yellow band variations (and hence the fouling rate) are minimal over the entire length of the channel at any point in time (differences between the upstream and downstream section are contained within 2% and 8.5%). Hence, it can be safely stated that fouling rates can be considered uniform over the membrane surface as inferred from filtrate flux measurements. These results are important from an experimental point of view for mainly two reasons. Firstly, they demonstrate that the fouling rates measured within the facility can be directly associated with quasi-uniform and hence well-definable flow conditions, hence paving the way to establish clear cause–effect relations between flow and mass-transfer processes. Secondly, thanks to uniformity, measurements performed by means of conventional flow/particle-measurement devices, which are local (or quasi-local as per PIV) in nature, can be considered representative of the whole system and hence directly relatable to integral quantities such as filtrate fluxes, without the need of lengthy and costly flow characterization over the entire unit. Figure 5b also provides some insights about the fouling behavior over time, which is clearly non-linear, as expected [31]. Indeed, variations in the yellow-band color were faster at the beginning of the experiment and slower towards the end. This observation is consistent with the data presented in Figure 6, representing the behavior in terms of filtrate fluxes vs. time (it should be noted that the initial flux rate is consistent with the values of permeance indicated by the manufacturer), which, as expected, displayed a faster drop during the initial part of the experiment and a slower reduction towards the end.

Initially, at higher flux value, colloidal deposition is faster and particles interacted with the membrane surface, possibly causing direct pore clogging, thus resulting in a higher flux reduction rate. Later during the tests, particle deposition slowed down due to a reduction in permeate flux (i.e., reduction in convective flow toward the membrane surface) and particles started depositing on a layer of pre-deposited particles, thus increasing the resistance of the cake layer but causing less direct pore clogging. More consistently with the objectives of the present study, Figure 6 provides evidence for the following important points:(i)Filtration tests carried out in the channel facility are controllable and repeatable because tests carried out at essentially identical experimental conditions display overlapping curves of filtrate vs. time (blue and green symbols).(ii)The pulsed flow case (red triangles, see Table 2) provided much improved filtrate flux behavior over time when compared to its steady flow counterpart (orange squares), as hypothesized in Section 3.2. Note that flux differences are much greater than those associated with experimental uncertainty, as supported by the previous point about experimental repeatability. Specifically, flux values were maintained always larger than 130 $L/m^{2}/h$ in the pulsed flow case and a final normalized flux of 0.9 was achieved, thus halving the fouling-related flux reduction compared to the stead flow case. This observation suggests the potential of hydrodynamic methods to reduce colloidal fouling in ultrafiltration membrane systems.

## 5. Concluding Remarks

In this work, we presented and validated a new module-scale lab facility designed to investigate interlinks between hydrodynamics and fouling in membrane flow systems. The facility is rather innovative as it allows for the development of turbulent flow conditions (often encountered in membrane systems) at an unprecedented quality and controllability at least when compared to past studies pertaining to the membrane science literature [32,33]. In this respect, PIV measurements indicated that fully developed turbulent channel flows were generated within the test section. These can also be considered canonical, in the sense that their velocity statistics compared very well with datasets presented in the literature and normally used as benchmark for quality check. Regarding filtration tests, these were carried out with waters containing a prescribed level of suspended colloids (clay) mimicking conditions of interest for the ultrafiltration of relevant surface waters. Results showed that tests were repeatable and displayed filtrate flux vs. time curves in agreement with those previously reported in the relevant literature. Despite the inherent (but slight) flow non-uniformity characterizing the filtration tests, fouling proved to occur rather uniformly along the whole unit. This was demonstrated from the measurements of filtrate fluxes at three different longitudinal sections of the unit, which proved to be nicely uniform and from the measurements of membrane color variations (as observed by means of an overhead camera and indicative of fouling), which also proved to be uniform. Finally, a pulsed flow test carried out by imposing pressure (and flow rate) variations at a dimensionless frequency, $\omega^{+}$ italics font removed of 0.0025, showed a significant reduction in colloidal fouling when compared to its steady counterpart, thus providing encouraging, albeit preliminary, results about pulsations as a viable hydrodynamic technique to reduce fouling in membrane systems. In this regard, ongoing work is focused to identify which pulsed-flow conditions, in terms of non-dimensional frequency and wave amplitudes, result in maximum fouling reductions within the context of applications in the water industry. 

## Figures and Tables

**Figure 1 membranes-12-00334-f001:**
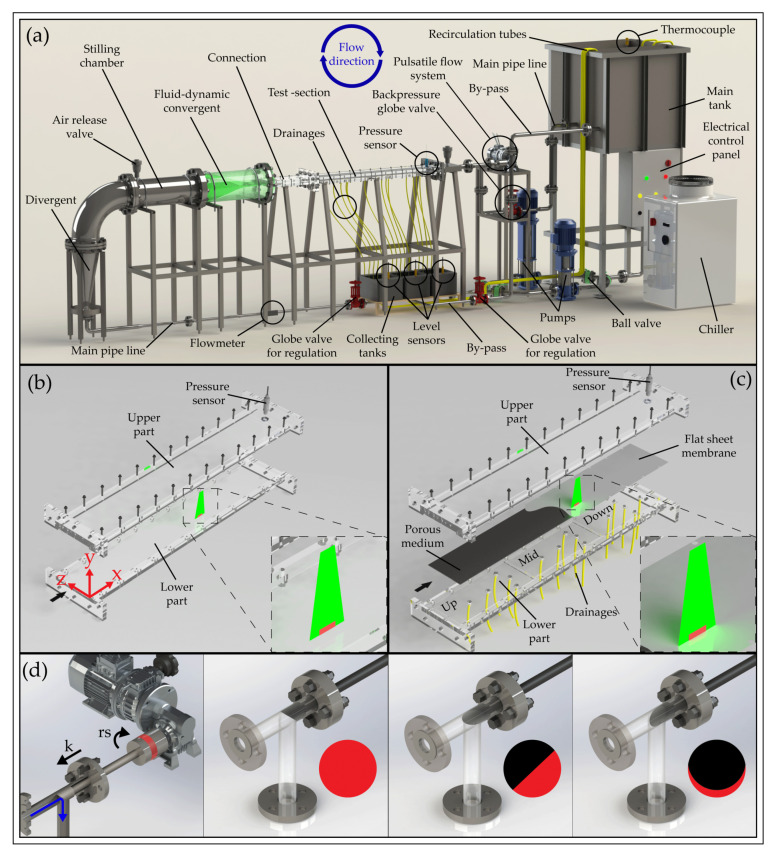
Overview of the proposed setup: (**a**) Rendering of the entire hydraulic facility. (**b**) Sketch of test section used for the fluid-dynamic validation of the system. (**c**) Sketch of the test section used for membrane tests. (**d**) Representation of the system used to generate the pulsating flow, where rs is the rotation speed and *k* is the distance of the rod in respect to the elbow. The outflow area (red) varies during each cycle as a function of the angle of the truncated rod (in this setup set at 45°). Panel (**b**) also shows the reference system used in the present study; the origin is chosen at the right-bottom entrance of the test section and x, y, z coordinates correspond to streamwise, wall-normal and spanwise directions, respectively.

**Figure 2 membranes-12-00334-f002:**
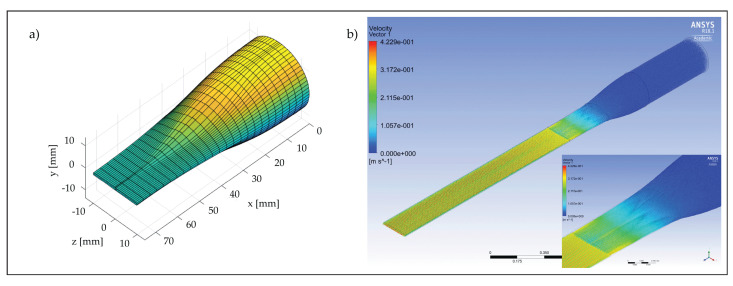
(**a**) Geometry of the convergent section and (**b**) result of a CFD simulation performed inside the fluid-dynamic convergent, connection and test section.

**Figure 3 membranes-12-00334-f003:**
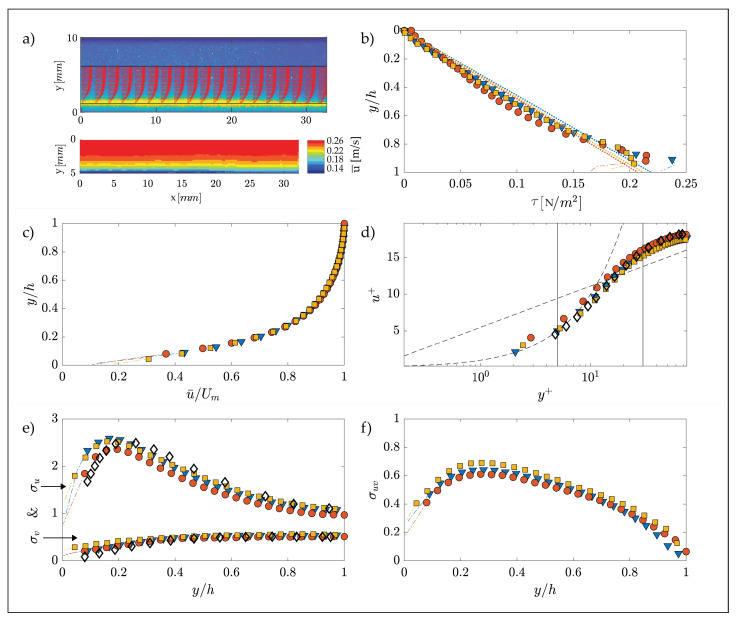
Velocity statistics obtained from PIV measurements in the lower half of the test section for three different streamwise positions; h is the channel semi-height, u${}^{+}=\bar{u}$/u* is the normalized velocity; $y^{+}$ = y u*/ν is the normalized distance from the wall, U*m* is the maximum speed, $\sigma_{v}=\sqrt{\bar{v^{\prime 2}}}$/u* and $\sigma_{u}=\sqrt{\bar{u^{\prime 2}}}$/u* are the non-dimensional root squares of the variance for wall-normal and longitudinal velocity fluctuations, respectively, and $\sigma_{uv}=\sqrt{\bar{v^{\prime }u^{\prime }}}$/u* is the non-dimensional root square of the covariance. (**a**) Flow field and contour plot for test F-1; (**b**) total shear stress for all tests; (**c**) non-dimensional mean-velocity profiles (outer scaling); (**d**) non-dimensional mean-velocity profiles (inner scaling), the dashed line represent the linear law u${}^{+}$ = y${}^{+}$, characterizing wall flows for $y^{+}$≲5 (see e.g., Pope (2000) [30]); (**e**,**f**) non-dimensional profiles of velocity variances. Symbols and experimental conditions for all the tests are reported in Table 1, with the exception for ◇ that refers to the data from the DNS by Tsukara (2005).

**Figure 4 membranes-12-00334-f004:**
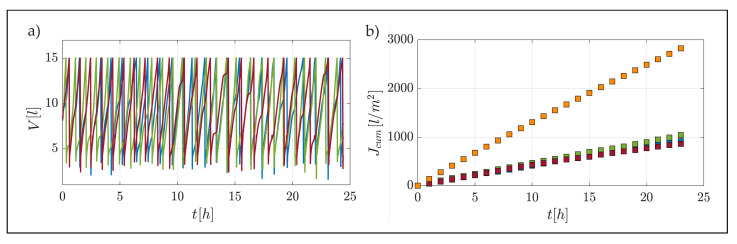
Results of S-4_1.5 filtration test with a feed pressure of 1.5 bar; (**a**) fill levels of the three filtrate collection tanks vs. time; (**b**) total water flux and flux measured from the individual portions. In both panels, blue, green and red lines/symbols refer to, up, mid and down collection tanks, respectively (see Figure 1). Orange squares in panel (**b**) refer to the cumulative flux.

**Figure 5 membranes-12-00334-f005:**
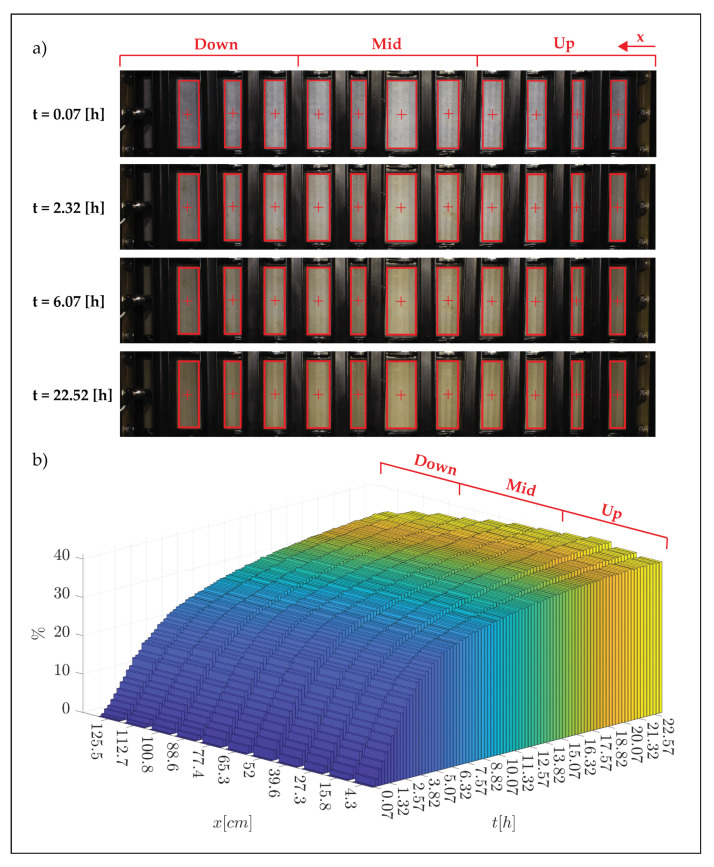
Data related to colloidal fouling of the membrane. (**a**) Photographs from the overhead camera taken over the entire span of the test section and at different times, t (test S-4_1.5); (**b**) percentage variation of the membrane color (yellow band) measured at different longitudinal positions along the membrane surface (i.e., in each of the red boxes highlighted in panel (**a**) as a function of time).

**Figure 6 membranes-12-00334-f006:**
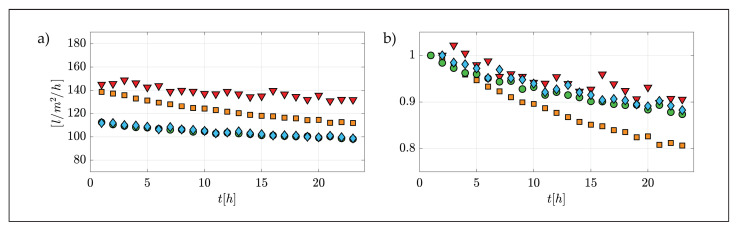
Dimensional (panel (**a**)) and non-dimensional (panel (**b**)) filtrate fluxes vs. time for all experimental conditions (symbols as indicated in Table 2). In Panel (**b**), filtrate fluxes are normalized by their initial value.

**Table 1 membranes-12-00334-t001:** Summary of the chosen hydraulic conditions to perform the flow-dynamic tests. The columns indicates: the Reynolds number, Re = Uh/ν, the viscous Reynolds number, $Re^{*}$ = u*h/ν, the maximum speed, $U_{m}$ and the friction velocity, u* =$\sqrt{\tau_{0}/\rho }$. For all tests, 2000 image pairs (window size 34.5 mm × 5 mm) were analyzed with a grid step size of 16 × 16 pixels and a spatial accuracy of 0.19 × 0.19 mm.

Test Name	Symbol	Coordinates [cm]	Re [-]	Re* [-]	$U_{m}$ [m/s]	$u^{*}$ [m/s]
F-1	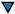	x = 82.5, z = 5	3981	70	0.267	0.0148
F-2	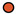	x = 34.5, z = 5	3981	70	0.267	0.0144
F-3	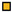	x = 130.5, z = 5	4038	68	0.265	0.0145

**Table 2 membranes-12-00334-t002:** Provides a summary of all experimental conditions for filtration tests. The columns indicate: the Reynolds number, Re = Uh/ν, the pressure, *P*, the suction coefficient, Γ = v/U and the non-dimensional frequency, $\omega^{+}=2\pi$ f ν/u*^2^.

Test Name	Symbol	Re [-]	*P* [bar]	Γ [$10^{-4}$]	$\omega^{+}$ [-]
S-4_1	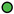	4005	1.00	1.37	-
S-4_1r	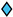	4013	1.00	1.38	-
S-4_1.5	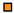	3983	1.52	1.63	-
P-4_1.5_2.5	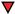	3974	1.51	1.84	0.0025

## Data Availability

The data presented in this study are available on request from the corresponding author.

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
