# Peer review of "New Facility for Membrane Fouling Investigations under Customizable Hydrodynamics: Validation and Preliminary Experiments with Pulsating Cross-Flow"

_membranes, 2022, doi:10.3390/membranes12030334_

Round 1
Reviewer 1 Report
The authors have submitted a well-written document that can be published in the current form.
Reviewer 2 Report
This work investigates a new module-scale laboratory facility that allows to control well the flow conditions and mass transport phenomena at the feed/membrane interface. They used particle image velocimetry measurements to confirm the uniform flow control. This work might help in designing filtration setup. The following are my concerns:
- What is the feed composition in this work? Feed property (viscosity, concentration, type of feed) affects the flow.
- Can the author draw the piping and instrumentation diagram (P&ID) clearly? This will help the reader to clearly understand the process.
- The equipment sizes like pipes and pump characteristics or tank size, it is suggested to provide in table form.
- What are the foulants used in this work? Other foulant have different behaviour in the membrane?
- What are the characteristic of the membrane used in this work? (surface charge, chemical functional group, or type of membrane, etc)
Reviewer 3 Report
This paper reports a method to reduce membrane fouling by hydrodynamic perturbations. A new module-scale lab facility was designed. It is confirmed that pulsating flow can reduce membrane fouling. However, the experimental results do not support the conclusions very well, and it is recommended to be rejected. Some comments are shown as follows:
Comment 1: Please carefully consider the logical order of the manuscript, and rearrange the figures. It is recommended to cut keywords to highlight the key points. The unnecessary introductory words and figures in the manuscript are also recommended to be cut, such as line 60-67 and Figure 2.
Comment 2: The amount of data is insufficient and the research depth is limited.
- In Abstract, it is mentioned that “Towards this end, in this work we propose and validate a new module-scale laboratory facility which allows to investigate, at very well-controlled flow conditions, how hydrodynamics affects mass transport phenomena at the feed/membrane interface.” However, in the body of the manuscript, you just proved that hydrodynamics can affect mass transport phenomena at the feed/membrane interface, not how. It is recommended to modify the expression.
- In 4.2. Filtration tests (line 326), it is recommended to attach data to confirm “the other tests showed a similar behavior”.
- In 5. Concluding remarks (line 377), it is recommended to attach references to support your conclusions.
Comment 3: It is recommended to carefully select experimental conditions.
- In 3.1. Fluid dynamic tests, more explanation is needed to illustrate why you choose the three experimental conditions.
- In 3.2. Filtration tests (line 233), except for total suspended solids concentrations and particle composition, are there other components that have impacts on experimental results?
Comment 4: Some figures and explanatory texts in this manuscript are not clear enough. It is recommended to label the legend in the figures rather than in the texts. Please attach axis names and write units by standard to avoid misreading. Figure 5 is too small to be understand and the notes below Figure 6 are difficult to read, please standardize the expressions.
Reviewer 4 Report
This paper reports on the design and implementation of a test set up that allows in detail investigation of the interplay between hydrodynamic conditions and fouling in membrane filtration. The success has ben validated by PIV characterization of the flow conditions in the system and by filtration experiments under steady and pulsed flow conditions. The results indicate that the system may be useful to gain fundamentally relevant insights and that well-defined / -controlled hydrodynamic conditions can have a major influence on fouling. The paper is also very well written. Overall, I have only a few comments.
The aim was to obtain fully developed and uniform flow inside the test section, and this has been validated by the experiments. However, the experiments have only been performed for one kind of hydrodynamic condition, characterized by Re ~ 4000. This raises several questions:
- The authors mention that they will mimic conditions of interest for ultrafiltration of relevant surface waters (cf. line 383); what are relevant applications performed at Re ~ 4000?
- In l. 306, it is written that result demonstrate “that the flows conform to canonical, turbulent channel flows”. I learnt that Re ~ 4000 corresponds to the transition region between laminar to turbulent. It there any concrete proof for turbulence? That could be obtained from more detailed analysis of PIV data.
3. Is it possible to obtain also well-developed flow conditions at lower velocities?
Minor points:
l. 40: “Jaffrin’s” should be “Jaffrin”
legend to Fig. 2: should read “convergent section”
Round 2
Reviewer 2 Report
all my concerns are addressed well.
Reviewer 3 Report
The revised manuscript is satisfactory.